# The Immunomodulatory Effect of *Nigella sativa*

**DOI:** 10.3390/antiox12071340

**Published:** 2023-06-24

**Authors:** Klaudia Ciesielska-Figlon, Karolina Wojciechowicz, Anna Wardowska, Katarzyna Aleksandra Lisowska

**Affiliations:** Department of Physiopathology, Faculty of Medicine, Medical University of Gdańsk, 80-210 Gdańsk, Poland

**Keywords:** *Nigella sativa*, essential oil, immunomodulation, antioxidant

## Abstract

Background: For thousands of years till nowadays, *Nigella sativa* (NS) has served as a common spice and food preservative. Its seed extracts, seed oil, and essential oil in traditional medicine have been used to remedy many ailments such as headaches, fever, gastric complaints, and even rheumatism. In addition, the antibacterial, virucidal, fungicidal, and antiparasitic properties of NS are well known. However, studies on the possible immunomodulatory effects of black cumin are relatively scarce. This article discusses in vitro and in vivo research supporting the immunomodulatory role of NS. Methods: The review is based on articles, books, and conference papers printed until September 2022, found in the Web of Science, PubMed, Wiley Online Library, and Google Scholar databases. Results: Experimental findings were reported concerning the ability of NS to modulate inflammation and immune responses or cytotoxic activity. Conclusions: All results suggest that NS can potentially be employed in developing effective therapeutic agents for regulating immune reactions.

## 1. Introduction

The long-standing interest in natural products has led to the emergence of plant substances in medicine. This branch of medicine is growing rapidly due to the increased effectiveness of new medicines of plant origin and the development of alternative therapies using plant products as support for traditional therapies for the treatment of various diseases. The use of plants as medicinal substances is culturally well-established and dates back to the earliest years of human activity [1]. It is known that people of different cultures have used the same plants for similar health problems, which have been repeatedly written down and passed down from generation to generation. Today, taking advantage of the possibilities of science, many of these plants have been studied, and substances isolated from them have shown beneficial therapeutic effects, including anticancer, antimicrobial, antioxidant, anti-inflammatory, and immunomodulatory effects. In the present article, we focused on the immunomodulatory potential of NS, describing studies using extracts, fixed and essential oil obtained from black seeds in vitro and in vivo models. We traced the articles published between 1995 and 2022. The literature included studies using animals, humans, and cell lines.

*Nigella sativa* (NS), a flowering plant that belongs to the Ranunculaceae family originates from south and southwest Asia. Presently, it is cultivated in several other regions including middle Europe, Mediterranean countries, and western Asia [2]. The plant grows up to a height of 20–30 cm and has linear leaves that are finely divided. Its flowers typically have 5–10 petals and come in pale blue, pink, or white colors, and bloom from May to September [3]. The fruit of the plant is a large and inflated capsule with three to seven united follicles, each containing seeds. The seeds are obtained when the plant reaches full maturity, which usually occurs in mid-August. They are small, black, and velvety, measuring 1–5 mm in size, and have a pungent taste resembling pepper, with a spicy aroma. The seeds are often used to season tender meats such as lamb and poultry and are also a great addition to salads, marinades, and cottage cheese [4]. In addition, cumin seeds are also used in bread and dairy products [5].

Seeds and oil from *Nigella sativa* have long been used in medicine and cooking [6]. This plant called the “gold of the Pharaohs”, was considered a panacea for many diseases in antiquity. Additionally, many ancient cultures, especially in Asia, used NS oil for the treatment of various allergies [7]. In Islamic countries, NS is extensively used in traditional medicine for healing numerous gastrointestinal and respiratory diseases [8]. In traditional and alternative medicine, NS is considered a panacea for various conditions, such as diarrhea and asthma [9]. Other uses include headache, amenorrhea, anorexia, cough, rheumatism, eczema, bronchitis, fever, influenza, as a diuretic, lactagougue, vermifuge, and various health care issues [10,11,12,13,14].

### 1.1. Composition of Seeds from NS

NS seeds contain proteins (26%), fat (28%), carbohydrates (25%), crude fiber (8.4%), and ash (4.8%) [15]. High levels of carotene and minerals (copper, zinc, phosphorous, and iron) are also found [15]. The high content of macromolecules, such as proteins, is crucial for the plant′s properties. Worth mentioning is a group of defense proteins, such as thionins and defensins, which are intracellular low-molecular proteins. Their beneficial properties are described by some studies, focusing on antibacterial or antifungal effects [16,17] and are even antiproliferative and pro-apoptotic against cancerous cell lines [18]. Furthermore, seeds hold 36–38% fixed oil, alkaloids, and saponins [10,19,20]. The oil obtained from NS seeds contains fatty and essential oil fractions [21]. The percentage of component content depends on time, location, and method of harvesting. NS seeds and their oils generally have very low toxicity [22].

The oil contains a fatty oil rich in unsaturated fatty acids, mostly linoleic acid (50–60%), oleic acid (20%), dihomolinoleic acid (10%), and eicodadienoic acid (3%). Saturated fatty acids (palmitic and stearic acid) amount to about 30% [23,24,25]. The oil contains alkaloids (nigellicines and nigelledine), saponins, tocopherols, phytosterols, flavonoids, and essential oil (EO) (0.4–2.5%) [26]. Most health-promoting properties are mainly attributed to quinone constituents, with thymoquinone (TQ) as a more abundant compound, which is also likely to be involved in pharmaceutical properties [27].

EO is utilized in food (as flavoring), perfumes, and pharmaceuticals [28]. Numerous active NS essential oil (NSEO) compounds have been isolated, identified, and reported in many experiments. Gas chromatography and gas chromatography-mass spectrometry analysis of volatile oil resulted in the identification of bioactive compounds representing even 98% of NSEO total amount [5,29]. In terms of quantity, the main identified compounds were p-cymene (50–60%) [5,29,30] and α-thujene (15%). Thymoquinone, thymohydroquinone, dithymoquinone, and thymol were the key phenolic compounds detected [31]. Traces of the esters of saturated and unsaturated fatty acids were also identified in NSEO [19]. Two monoterpenoids, cis- and trans-4-methoxythujane, were found in the essential oil [5]. Four terpenoids: cis-sabinene hydrate methyl ether, trans-sabinene hydrate methyl ether, 1,2-epoxy-menth-4(8)-ene, and 1,2-epoxy-menth-4-ene were identified in NSEO by nuclear magnetic resonance [32].

Figure 1 shows the chemical structures of the most important compounds of *Nigella sativa*.

### 1.2. Antioxidant Activity of the Seeds NS Extracts

Many studies using different experimental models highlight the antioxidant activity of NS extracts. Bordoni et al. [33] verified fixed oil′s antioxidant properties in an in vitro model of inflamed adipose tissue using Simpson–Golabi–Behmel syndrome (SGBS) human preadipocytes and monocytic leukemia cell line. The authors used different spectrophotometric and chemiluminescent assays to measure the antioxidant of NS oil. The total antioxidant activity (TAA) measured in the supernatant of preadipocytes showed that NS oil has very high residual activity. Tiji et al. [34] compared the antioxidant properties of NS seed hexane and acetone extract and fractions. The authors confirmed that the extracts are characterized by the presence of fractions with different levels of antioxidant activity, which probably depends on specific secondary metabolites present in the fractions, such as polyphenols. Another study showed that methanolic extract from NS seeds has higher antioxidant activity than aqueous extract [35]. Ouattar et al. [36] showed that NS extracts have a dose-dependent antiradical activity, with crude extract having an activity lower than n-butanol or ethyl acetate extracts. As can be seen, NS exhibits antioxidant activity, which was confirmed by various methods. However, the level of this activity depends on the type of extract and the fraction tested.

### 1.3. The Most Efficient Methods for Obtaining and Detecting the Composition of Oils and Essences from Nigella sativa

Many extraction techniques have been developed to extract oils from *Nigella sativa* seeds. A relevant aspect worth mentioning is that the different forms of extraction of NS seed oils are one of the main factors influencing their final properties. Therefore, it is crucial to adapt the extraction technique to the specific application [37].

Some methods allow the isolation of macromolecules, such as proteins. In 2016, innovative research was conducted on the proteome of *Nigella sativa* seeds [38]. The researchers analyzed the protein profile of NS seeds with a proteome mapping technique using one-dimensional gel electrophoresis followed by liquid chromatography and tandem mass spectrometry strategies. Next, the vanillin method, for example, can be used to extract such compounds.

Cold extraction, also known as cold press, is among the conventional oil extraction methods. It does not require auxiliary chemicals, so it is preferred by those concerned about natural and safe foods [39]. However, this method is characterized by poor overall efficiency [40]. Additionally, the residual meal contains only about 11% of oil, which may severely limit usage in the food industry [40].

The hydrodistillation (HD) method is among the traditional techniques for extracting essential oils and volatile components together with steam distillation (SD). In this method, the hydrolysis temperature is essential for improving the borage oil extraction yield by cold pressing. Presently, HD is mainly used as a comparative method for newly developed methods for extracting volatile components of NS [5,40,41].

According to the literature, NS oil has often been produced by hot solvent extraction (SE) [42]. Among the most common solvents used to extract essential and solid oils from NS are hexane, chloroform, acetone, and methanol [5,20,40,43,44,45]. At the same time, its disadvantage is the requirement for high temperatures, which can degrade the desired components of the oils [46].

Considering the lack of selectivity of the previously mentioned methods due to insufficient control of the individual parameters of both techniques, a new supercritical extraction (SFE) method was developed [47]. Compared to other conventional techniques, higher yields of antioxidants, greater purity of the extracts, and more considerable retention of bioactivity are the main advantages of SFE. To date, many extractions of NS seed oils have been performed using the SFE method. Thus, in a study, Kokoska et al. [48] compared the chemical composition and antimicrobial activity of the essential oil extracted by the SFE-SD method and the traditional HD and SD methods. The results showed that the extract obtained by the SFE-SD method differed in composition and higher bioactivity compared to traditional methods. Additionally, a study performed by Ismail et al. [49] showed that NSO extracted using the SFE method contained significantly higher amounts of TQ than the oil extracted using the solvent extraction method.

Another technique is microwave-assisted extraction (MAE), belonging to alternative methods for extracting essential oils, aromatics, pesticides, phenols, and other organic compounds. Microwave heating offers the advantage of breaking weak hydrogen bonds promoted by the dipole rotation of the molecules [50]. A study by Benkaci-Ali et al. [51] showed that extracting essential oil from NS seeds using MAE had a shorter time to obtain the oil and significant energy savings compared to the HD technique.

Meanwhile, gas chromatography (GC) is among the most commonly used methods to identify and determine the chemical composition of essential oils and lipids obtained from the extraction of NS seeds. Using a combination of the GC-MS technique with iterative and non-iterative resolution methods, the known volatile components in NS seeds increased from 39 to 98 [52]. Interestingly, the GC technique has also been used to study differences in chemical composition between ancient and modern *Nigella sativa* seeds. As a result, it was shown that the composition of both seeds was very similar in the content of essential oils in the plants [53].

## 2. Immunomodulatory Properties of *Nigella sativa*

Immunomodulation is defined as modifying the immune response by regulating the crosstalk among different components, such as interactions between neutrophils and macrophages and T and B cells. Immunomodulators can help support immune function by stimulating or suppressing the immune system [54]. Below we discuss the available literature data regarding the immunomodulatory effect of NS seed extracts, fixed oil, and essential oil.

### 2.1. Immunomodulatory Properties of NS Seed Extracts

#### 2.1.1. Studies In Vitro

Haq and colleagues [55] examined the effect of phosphate buffer saline (PBS) extract from NS seeds in 50, 5, and 0.5 µg/mL concentrations on human lymphocytes and polymorphonuclear (PMNs) leukocytes. Peripheral blood mononuclear cells (PBMCs) were stimulated with concanavalin A (ConA), phytohemagglutinin (PHA), or pokeweed mitogen (PWM) in the presence of several concentrations of NS extract. The high concentrations of NS extract suppressed the lymphocyte response to all mitogens due to increased cell death. Furthermore, at the highest concentration, NS extract also suppressed the phagocytic activity of PMNs. At the same time, the extract stimulated lymphocytes to secrete interleukin 1beta (IL-1β) and IL-3 but not IL-2. Majdalawieh and colleagues [56] examined how NS influences splenocyte proliferation, macrophage function, and natural killer (NK) cell activity in C57/BL6 and BLAB/c cells. The authors demonstrated that the aqueous extract from NS seeds in four doses (1, 10, 50, and 100 g/mL) increased splenocyte proliferation and the secretion of Th2 cytokines responsible for humoral immune responses in a dose-dependent manner by splenocytes. At the same time, it suppressed the secretion of key proinflammatory mediators, that is, tumor necrosis factor-alpha (TNF-α), IL-6, and nitric oxide (NO) by macrophages. Finally, the authors showed that the extract significantly augmented NK cell cytotoxicity against YAC-1 tumor cells. This study showed that the effect of the NS extract depends on the type of immune cells, as some immune subpopulations were stimulated in the NS presence to proliferate and produce cytokines, while others were inhibited.

Hexane and methanol extracts from NS seeds in a concentration of 10 mg/mL were found to prevent the formation of protein carbonyl and depletion of glutathione (GSH) in L929 fibroblasts exposed to toluene, which confirms that NS has antioxidant potential [57]. In 2013, Elmowalid and colleagues [58] examined the immunomodulatory effect of an aqueous extract concentration of 10 mg/mL from NS seeds on sheep macrophage functions in vitro. The authors showed that the addition of NS extract caused an increase in phagocytic activity and the capacity to produce NO. In the study by Gholamnezhad et al. [59], ethanolic extract from NS seeds in concentrations of 100, 500, and 1000 µg/mL was examined to see whether it affected rat splenocytes, especially their viability, proliferation, and cytokine secretion. NS extract inhibited the proliferation and decreased the viability of splenocytes stimulated with PHA or ConA in a dose-dependent manner. Higher concentrations of the extract also reduced the secretion of IL-4 and interferon-gamma (IFN-γ) and increased the IFN-γ/IL-4 ratio. In a recent study, Singh et al. [60] investigated the immunomodulatory effect of methanolic extract from NS seed in two doses, 125 and 250 μg/mL, on chicken PBMCs. The authors demonstrated that the extract stimulated the expression of inflammatory genes coding IL-1β, IL-4, IL-10, IL-12, IL-13, IFN-β, and IFN-γ.

However, scarce studies on human material have been reported. For example, Koshak and colleagues [61] compared the effects of ethanol, aqueous, and supercritical fluid (SCF) extracts from NS seeds in concentrations of 10 and 100 μg/mL on asthma-related mediators of inflammation in human T cells and monocytes. SCF had the most robust suppressive properties; the extracts reduced the release of IL-2, IL-6, and prostaglandin E2 (PGE2) from T cells and monocytes. Alshatwi [62] showed that methanolic extract in concentrations of 2.5 and 5.0 μg/mL from NS seeds inhibited the proliferation of T cells stimulated with PHA for 48 h in a dose-dependent manner. However, without PHA stimulation, NS extract stimulated T cells. The NS extract also decreased the expression of genes coding TNF-α, IL-6, and IL-8 in human PBMCs stimulated with PHA. Conversely, in the absence of PHA stimulation, gene expression was increased in the presence of NS extract.

#### 2.1.2. Studies In Vivo

The in vivo immunomodulatory properties of extracts from NS seed were also examined in different animal models. For example, intraperitoneal administration of NS methanolic extract increased the total amount of white blood cells (WBC) in BALB/c mice [63]. In addition, hexane and methanol extracts prevented the loss of hepatic GSH in male Wistar rats exposed to toluene [57].

Boskabady and colleagues [64] studied the immunomodulatory properties of ethanolic extract from NS seeds in guinea pigs sensitized to ovalbumin (OVA) as an animal model of asthma. The authors showed that NS extract significantly decreased pathological changes in the lungs of animals. The infiltration of eosinophils and lymphocytes, as well as local epithelial necrosis, was reduced. Simultaneously, serum IL-4 and IFN-γ were increased in animals treated with NS extract compared to control animals. These results confirmed the preventive effect of NS extract on lung inflammation. In the latest study by Hikmah and colleagues [65], the influence of ethanol NS extract on renal tissue damage in the pristane-lupus (PIL) mice model has been investigated. The results showed that the percentage of Th17 cells responsible for inflammatory responses, regulatory T cells (Tregs), and macrophages producing IL-6 and IL-23 in lupus mice treated with NS extract was lower compared to PIL mice treated with placebo or steroids. Additionally, the serum anti-dsDNA antibodies were lower in these mice. Additionally, the renal injury was smaller in lupus mice treated with NS extract. These results showed that the ethanolic extract from NS seeds has immunomodulatory properties and prevents kidney injury in lupus mice.

In another study, the immunomodulatory properties of ethanol extract from NS seeds were assessed in dexamethasone-induced immune-suppressed male rabbits [66]. Animals were treated for six weeks orally with water, NS extract, and dexamethasone (Dex) in different combinations. The authors reported a significant decrease in the phagocytic activity of the cells in rabbits treated with Dex alone. However, the administration of NS extract simultaneously with Dex or after three weeks of treatment with Dex improved the phagocytic activity. The authors also showed that NS extract improved bone marrow mitotic activity after treatment with Dex.

The properties of NS were also studied in fish [67]. In 2012 Elkamel and Mosaad published work that investigated the modulation of the immune system of *Nile tilapia* by NS seeds added as a food additive by comparing its properties with fish basic or a CloSTAT diet (diet with the addition of Bacillus subtilis PB6). The authors investigated immune parameters, including serum globulins, WBC counts, and phagocytic activities. Results showed that adding NS seeds significantly increased the serum globulins, WBC count, and phagocytic activity and reduced fish mortality compared with the standard or CloSTAT diet. These effects were even significantly higher when fish were treated with NS seeds combined with the CloSTAT diet.

In another study, the prophylactic properties of water extract from NS seeds were studied in a group of asthmatic patients [68]. The study group received boiled NS seed extract for 3 months. Asthma severity and frequency of symptoms per week, and wheezing were analyzed three times: in the beginning, 45 days after treatment, and at the end of the study. All asthma symptoms, frequency of attacks, chest wheezing, and pulmonary function test (PFT) values significantly improved in the second and third visits compared with the beginning of the experiment.

### 2.2. Immunomodulatory Properties of NS Oil

#### 2.2.1. Studies In Vitro

Only a few studies describe the immunomodulatory properties of NS cold-pressed oil in vitro. In 2005, Buyukozturk et al. [69] examined how NS oil influences cytokine production by splenic mononuclear cells (MNCs) in mice. First, BALB/c mice have been give daily 0.3 mL of NS oil through an oro-esophageal cannula for a month. Then, in the third week of the study, all mice were sensitized using intraperitoneal injections of 20 µg of OVA. Finally, splenic MNCs from mice were cultured with OVA or ConA. The authors showed that the cytokine production did not significantly differ between mice treated with NS and control mice treated only with saline solution. In the second study, the authors examined the influence of NS oil on human PBMCs stimulated with an immobilized monoclonal anti-CD3 antibody in the presence of serial (1:1, 1:10, 1:100, and 1:1000) ethanol (EtOH) dilutions of NS oil [70]. The lowest dilutions (1:10 and 1:50) of NS oil inhibited the proliferation of lymphocytes and reduced the percentage of living cells by inducing apoptosis.

#### 2.2.2. Studies In Vivo

More studies demonstrate the influence of NS oil in vivo in animals and humans [71]. In one of the studies, the authors studied the influence of NS oil in a rat model of allergic airway inflammation. Rats were first treated with intraperitoneally administered oil and then exposed to OVA. The authors measured serum levels of total immunoglobulin E (IgE), IgG1, and OVA-specific IgG1 and analyzed the proliferation of T cells in the spleen. They also analyzed the expression of genes coding different cytokines, including IL-4, IL-5, IL-6, and transforming growth factor-beta 1 (TGF-β1). The results showed that NS oil suppressed the Th2-type response in rats by preventing inflammatory cell infiltration and pathological lung lesion formation. NS oil also significantly decreased NO production in bronchoalveolar lavage fluid (BALF), total serum IgE, IgG1, OVA-specific IgG1, and IL-4, IL-5, IL-6, and TGF-β1 gene expression. Treatment with NS oil also inhibited the proliferation of T cells in the spleen. Balaha and colleagues [72] investigated the anti-inflammatory and immunomodulatory effects of oral administration of NS oil in a mouse model of allergic asthma. The authors examined the airway function, the presence of inflammatory infiltrates in the airways, local cytokine production in BALF, serum immunoglobulin concentrations, and histopathological changes in lung tissues. They showed that oral treatment with NS oil at a dose of 4 mL/kg/day in OVA-sensitized mice significantly improved airway reactivity, decreased the number of WBC, macrophages, and eosinophils, reduced the production of Th2 cytokine (IL-4, IL-5, and IL-13), and significantly increased IFN-γ with the abrogation of histological changes in lung tissues in a dose-dependent manner. Sheir and colleagues in 2015 [73] examined the immune mechanisms possibly involved in ameliorating histopathological changes in the livers of mice infected with *Schistosoma mansoni* and treated with NS oil. The authors measured total serum IgG and cytokines IL-2, IL-12, and TNF-α. Mice treated with NS oil were characterized by increased serum IgG, IL-2, IL-12, and TNF-α compared with infected mice. Additionally, histological observation of liver tissue of infected mice treated with NS oil combined with antiparasitic drugs showed some improvement compared with infected mice.

The authors of another study examined the influence of NS oil on CD4+ T cells in Sprague-Dawley rats exposed to dimethylbenzanthracene (DMBA), a carcinogenic immunosuppressive agent [74]. Animals were fed different doses of NS oil for 14 days before and during DMBA induction. At week 27, peripheral blood was taken to measure the number of CD4+ and CD4+ CD25+ T cells. NS oil administration increased the absolute number of CD4+ T cells and CD4+ CD25+ cells (Tregs) count in a dose-independent manner.

It has been shown that NS cold-pressed oil also influences the human immune system. In 2016 Kheirouri et al. [75] published a study that investigated the immunomodulatory effect of NS oil on selected subsets of T cells in women with rheumatoid arthritis (RA). Forty-three female RA patients were recruited and received 1 g of NS oil or starch capsule (placebo) in two dosages for two months. The Disease Activity Scores in 28 joints (DAS28) were calculated, and percentages of blood CD4+, and CD8+ T cells were analyzed. Treatment with NS oil significantly decreased the serum concentration of high-sensitivity C-reactive protein (hs-CRP) and reduced the DAS-28 score. Patients also reported a reduction in the number of swollen joints. No changes were observed in the percentage of CD4+ T cells before and after treatment. NS oil administration reduced the percentage of CD8+ T cells and increased CD4+CD25+ T cells and the CD4+/CD8+ ratio compared to RA patients treated with placebo and baseline results. Another study showed that NS oil affects Th1/Th2 cytokine balance and improves asthma control in children [76]. Children in this study were given 15–30 mg/kg/day of NS oil for eight weeks. The asthma control test (ACT) score was used to assess the improvement of asthma control. Numbers of blood Th1 and Th2 cells and serum IFN-γ and IL-4 were analyzed. The authors demonstrated that children treated with NS oil had significantly elevated serum IFN-γ and reduced IL-4 compared with the placebo group. However, the ACT score was not significantly different between groups at the end of the study. In another trial, the benefits of NS oil supplementation in asthma patients were studied [77]. Each participant received capsules of 500 mg twice a day for 4 weeks. Compared with the placebo, patients supplemented with NS showed a significant improvement in mean ACT score and a significant reduction in blood eosinophils. Hidayati et al. [78] analyzed the immunomodulatory properties of NS oil in active smoker volunteers. The authors showed that people receiving 3 × 1, 3 × 2, and 3 × 3 capsules/day for 30 days were characterized by increased IL-2 expression in CD4+ cells. Laily et al. [79] examined the effect of NS oil in a smoking group on serum IL-1β levels and neutrophil percentage. The study group received three-dose NS oil for 30 days. The authors showed that the average IL-1β levels did not differ between placebo and study groups. The neutrophil percentage, however, decreased after 30 days compared with the placebo group.

### 2.3. Immunomodulatory Properties of NS Essential Oil

#### 2.3.1. Studies In Vitro

Only one study examines the immunomodulatory properties of NS essential oil (NSEO) in vitro [29]. The authors stimulated human PMBCs with an immobilized monoclonal anti-CD3 antibody in the presence of serial (1:10, 1:50, 1:100, 1:500, or 1:1000) EtOH dilutions of NSEO. Their results showed that 1:10, 1:50, and 1:100 NSEO strongly inhibited the proliferation of CD4+ and CD8+ T cells and induced their apoptosis and necrosis in a dose-dependent manner. Additionally, the authors observed reduced surface expression of CD28 and CD25 antigens, which are essential for lymphocyte activation. The results obtained explained the immunomodulating effect of NS cold-pressed oil observed earlier by the authors.

#### 2.3.2. Studies In Vivo

In 2004, Nazrul Islam et al. [80] published a study that examined the immunosuppressive and cytotoxic properties of NSEO in a rat model. The authors challenged Long-Evans rats with typhoid antigen and injected them with NSEO for 30 days; then analyzed peripheral immune cells (lymphocytes, monocytes, neutrophils, and eosinophils) and serum immunoglobulins. NSEO significantly decreased neutrophil counts and increased lymphocytes and monocytes in the experimental animals. Meanwhile, serum immunoglobulins were decreased in animals treated with NSEO.

Figure 2 shows the potential immunomodulatory effect of *Nigella sativa,* and Table 1 summarizes its known immunomodulatory properties.

## 3. Conclusions

Since the 1950s, the phytochemicals and therapeutic properties of *Nigella sativa* have been studied exhaustively. Its antibacterial, antiviral, and antifungal properties are well-known and widely described, but we know little about its effects on the immune system. The research results described in this review demonstrate that the NSEO influences immune cells by modulating their parameters such as the ability to produce cytokines or nitric oxide, phagocytic and cytotoxic activity, splenocyte and T-cell proliferation, and susceptibility to apoptosis and necrosis both in vivo and in vitro. Therefore, NS is a promising source of active ingredients that could be implemented in different clinical settings. However, further studies are necessary to explore the mechanisms of cold-pressed oil and essential oil from *Nigella sativa*, particularly in humans at the cellular and molecular levels. This review could be considered a compass for upcoming studies and the future development of therapeutic agents to help reduce symptoms of immune-related diseases, such as asthma or rheumatoid arthritis.

## Figures and Tables

**Figure 1 antioxidants-12-01340-f001:**
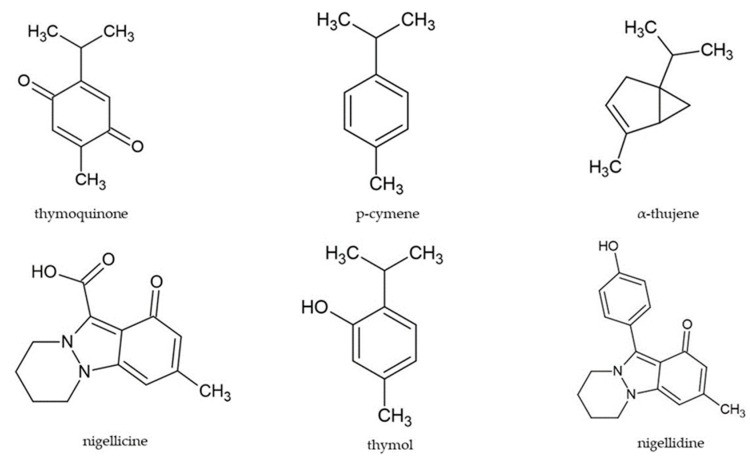
The chemical structures of the main compounds of *Nigella sativa*.

**Figure 2 antioxidants-12-01340-f002:**
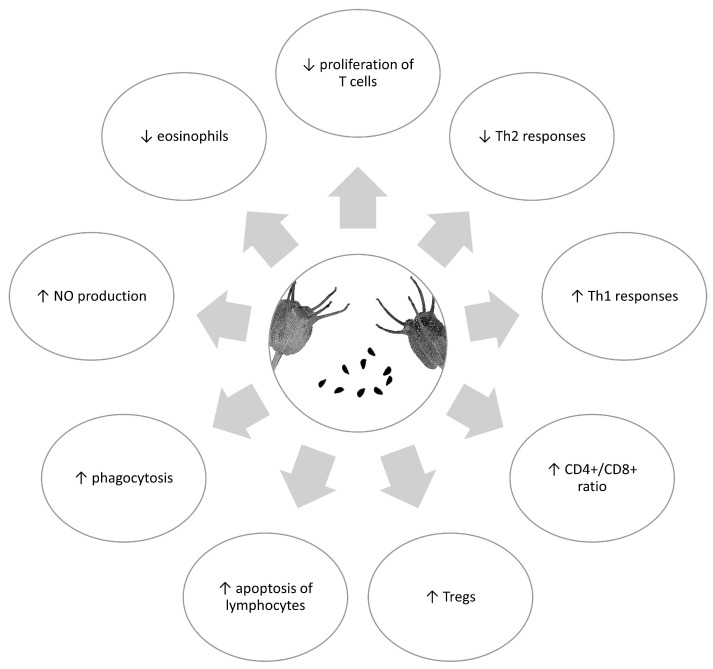
Potential immunomodulatory effects of *Nigella sativa*.

**Table 1 antioxidants-12-01340-t001:** The summary of the immunomodulatory properties of *Nigella sativa*.

**Type of Material**	**In Vitro**	**In Vivo**
Seed extracts	- PBS extract of 50 µg/mL NS induces cell death of human PBMCs, stimulates IL-1β and IL-3 secretion, and suppresses the phagocytic activity of PMNs [55] - aqueous extract in concentrations 1, 10, 50, and 100 g/mL of NS seeds increases splenocyte proliferation and secretion of Th2 cytokines, inhibits the secretion of IL-6, TNF-α, and NO, and increases the cytotoxic activity of NK cells against YAC-1 tumor cells [56] - the aqueous extract in a concentration of 10 mg/mL increases phagocytic activity and the NO production capacity of macrophages [58] - the ethanolic extracts in concentrations of 500 and 1000 µg/mL reduce the viability and inhibit the proliferation of rat splenocytes [59] - methanolic extracts in two doses of 125 and 250 µg/mL stimulate the expression of genes coding IL-1β, IL-4, IL-10, IL-12, IL-13, IFN-β, and IFN-γ in chicken PBMCs [60] - SCF extract in concentrations 10 and 100 μg/mL reduces the release of IL-2, IL-6, and PGE2 from human T cells and monocytes [61] - methanolic extracts in concentrations of 2.5 and 5.0 μg/mL inhibit human T cell proliferation and reduce the expression of genes encoding IL-6, IL-8, and TNF-α [62]	- the methanolic extract increases total WBC count in BALB/c mice [63] - the ethanolic extract reduces lung infiltration of eosinophils and lymphocytes and increases serum IL-4 and IFN-γ in OVA-sensitized guinea pigs [64] - the ethanolic extract reduces kidney damage, percentages of Th17 and macrophages in PIL mice [65] - the ethanolic extract improves bone marrow mitotic activity in rabbits with Dex-induced immunodeficiency [66] - seeds increase WBC count and phagocytic activity in *Nile tilapia* [67] - the aqueous extract reduces clinical symptoms of asthma patients [68]
Fixed oil	- oil in ethanolic dilutions 1:1, 1:10 inhibits human T cell proliferation and induces their apoptosis [70]	- oil inhibits Th2-type response and prevents allergic airway inflammation in OVA-sensitized rats [71] - the oil improves airway reactivity, decreases blood WBC, macrophages, and eosinophils, inhibits Th2 cytokine production, and increases IFN-γ in OVA-sensitized mice [72] - oil increases serum IL-2, IL-12, and TNF-α in mice infected with Schistosoma mansoni [73] - oil increases CD4+ T cell and Tregs counts in Sprague-Dawley rats exposed to DMBA [74] - oil reduces serum hs-CRP levels, DAS-28 scores, and the number of swollen joints in RA patients; it also reduces the percentage of CD8+ T cells and increases CD4+CD25+ T cells and the CD4+/CD8+ ratio [75] - oil reduces serum IL-4 and increases IFN-γ in asthma children [76] - the oil improves ACT scores and reduces eosinophil counts in adult patients with asthma [77] - oil increases IL-2 gene expression in CD4+ T cells and decreases neutrophil percentage in smokers [78]
Essential oil	- EO in ethanolic dilutions 1:10, 1:50, and 1:100 inhibits the proliferation of CD4+ and CD8+ cells, induces their apoptosis, and decreases the expression of CD28 and CD25 antigens [29]	- EO induced a significant decrease in neutrophil counts and increased lymphocytes and monocytes in rats exposed to typhoid antigen [80]

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
