# Peer review of "The Immunomodulatory Effect of Nigella sativa"

_antioxidants, 2023, doi:10.3390/antiox12071340_

Round 1

Reviewer 1 Report

 The immunomodulatory effect of Nigella sativa

Authors : Klaudia Ciesielska-Figlon, Karolina Wojciechowicz, Anna Wardowska and Katarzyna A. Lisowska

Manuscript# 2431240 peer review v1

Summary

This manuscript reviews current knowledge concerning the immunomodulatory effects of Nigella Sativa, used for millennia as natural therapeutic and medicinal substances and food flavourings. This is an important area of research as fossil-fuel sources, the feedstocks for bulk industrial chemical synthesis of common pharmaceuticals recede. Increasingly, the interrelationship between our immense knowledge to date of physics, chemistry, biochemistry and molecular biology seems the perfect partnership for expanding the field of plant-based therapeutics as a sustainable avenue for the future and long-term health of the human race.

I commend the authors for providing a sound platform for future research in this particular area.

Overall Opinion

The presentation of this manuscript is exemplary - well organised, concise and a delight to review. Excellent English, good grammar and descriptive writing. I recommend this manuscript for publication, with only minor comments.

Specific comments

Lines 47/48 Repetition, Remove the second sentence ‘NS seeds are characterised by a spicy smell.’

Line 57  grammar it should read ‘…used blackseed oil for treatment of various allergies [7].4

Line 60  spelling I prefer ‘diarrhoea’

Line 61 spelling again I prefer ‘amenorrhoea’

Line 78 grammar Quinonic alkaloids are also likely to be involved in pharmaceutical properties [24].

Line 85/86  grammar ‘….and thymol were the major phenolic compounds detected’

Line 175 grammar ‘….confirms that NS has antioxidant potential [49].’ Delete ‘the’

Line 190 grammar ‘…human material has also been reported.’

Line 198 grammar The NS extract also…..’

Line 269 definition have you defined what OVA means by this stage ? Ontario Volleyball Association? Organic Vapour Analyser?

Line 274 grammar should read ‘lesion’ rather than ‘lesions’

Lines 283 & 233 definition have you defined WBC? I assume white blood cell count

Line 338 grammar ‘ …. the results obtained’

Observations

1.       Would it be beneficial to have a table position around 1.3 to 1.4 which shows the chemical structures of the most important compounds.

2.       While I like both Figure 1 and Table 1, should the entries in Table 1 have the relevant reference number/s attached to each entry?

References

I have checked enough references, randomly selected, to know that the citation detail is correct. My only question for the editor concerns the layout. Should the Initials be punctuated?? For example reference 4, should it remain Yimer, EM;  or should it be Yimer, E.M.; …..Personally I prefer punctuated.

Summary

I will recommend this manuscript for publication, but after these minor details have been addressed. I will inform the Editor that I do not need to see the manuscript resubmitted – just the tiny details changed.

Excellent presentation, really good English. Well done to everybody invloved with the presentation of the manuscript. 

Reviewer 2 Report

This review is a very good work that consolidates all the experimental data on the effects of black cumin (Nigella sativa) on the human and animal immune systems. At the same time, the results are systematized separately for in vitro and in vivo experiments, including for various so-called physical substances of black cumin, namely, various extracts of seeds, fatty and essential oils. I am fully convinced that the information presented in this work on the molecular mechanisms of regulation of the human and animal immune systems in response to the application of N. sativa will help explain a number of therapeutic effects shown in numerous publications devoted to this plant which have been publishing for a long time.

As mentioned earlier, the article makes an extremely favorable impression, but there are a number of key points that it is very desirable to reflect in the work:

1. Sections 1.1. and 1.2 should be merged into one and transferred to “Introduction”. Here are fairly well-known points that are not directly related to the topic of the review. In particular, in section 1.2, the authors list many diseases for which NS is used. Such a list can be significantly expanded; there is no connection between immunomodulation and the achievement of a therapeutic effect.

2. Section 1.3 should be supplemented with known data on the content of many biologically active macromolecules, mainly proteins and peptides, in NS seeds. It is extremely incorrect to associate the variety of effects of the plant in question solely in terms of small molecules. There are still a lot of questions in this direction, but a beginning has already been made.

3. Section 1.4. must be supplemented with a methodology for the isolation of biologically active macromolecules in accordance with the new information requested in section 1.3.

4. Section 2, subsection 2.1 - the authors present literature data on the immunomodulating effect of NS extracts, and in most cases we are talking about water, water/salt and water/organic extracts. It is quite obvious that in such variants the active components are polar/hydrophobic small molecules and amphiphilic macromolecules, not at all substances of a quinone nature. How can authors comment the presence of activity?

5. Section 2.2, subsection 2.2.1 and section 2.3, subsections 2.3.1/2.3.2. – the authors provide a few results on in vitro immunomodulatory activity for NS oil and in vitro/in vivo for NS essential oils. What could be the reason for this, according to the authors? And on what basis are the in vivo results for NS oil based in this case?

Reviewer 3 Report

Comments to the Manuscript ID: antioxidants-2431240

Title: The immunomodulatory effect of Nigella sativa

This manuscript is a review on the different uses of Nigella sativa as food preservative and in the traditional medicine, for different applications. The authors reported the results of studies on the immunomodulatory effects of Its seed extracts, seed oil and essential oil, both in vitro and in vivo.

-       The manuscript is in line with the aim of Antioxidants, however, some corrections are needed.

-       First of all, in the key words list the word: antioxidant, can be added.

-       In the paragraph 1.1 Nigella sativa botanical characteristics and widespread usage plagiarism has been detected.

-       Paragraph 1.2: In traditional and alternative medicine, NS is considered a panacea for various conditions, such as diarrhea and asthma. Other uses include amenorrhea, anorexia, asthma, cough, rheumatism, bronchitis, headache, fever, influenza, eczema, as a diuretic, lactagougue, vermifuge, and various health care issues. Line 61: delete the word “asthma” as it was previously cited, so it is not another used in the following list.

-       After the paragraph: 1.3. Composition of seeds from NS it would be useful to add a paragraph on the “Antioxidant activity” of the seeds NS extracts. In the text the antioxidant activity if NS extracts is cited, but it is not considered in detail, I think that just a paragraph on it is needed.

-       Paragraph 2 Immunomodulatory properties of Nigella sativa

-       The authors report the results cited in this paragraph in table 1 altogether, at the end.

-       Insert at least 1 table for seed extracts, 1 table for oil and 1 table for essential oil to summarize the results of the reported references. In this way it would be clearer to follow what reported in the text, instead to look at only one large table at the end.

-       References: Although as the authors wrote, further work must be done on this subject, more recent references are needed at least in the period 2018-2022.

The quality of the English is quite good.

Round 2

Reviewer 2 Report

The authors have answered all my key points. There are no additional comments here.